# Mississippi Small Farm Product Amounts, Seasonality, and Proximity to K-12 Public Schools

**DOI:** 10.3390/ijerph20043572

**Published:** 2023-02-17

**Authors:** Jessica L. Thomson, Alicia S. Landry, Tameka I. Walls, Randall McMillen

**Affiliations:** 1US Department of Agriculture, Agricultural Research Service, Stoneville, MS 38776, USA; 2Department of Nutrition and Family Sciences, University of Central Arkansas, Conway, AR 72035, USA; 3Geosystems Research Institute, Mississippi State University, Starkville, MS 39759, USA

**Keywords:** small farms, public schools, Mississippi, farm to school, online survey, local foods, spatial analysis

## Abstract

The study’s purpose was to determine small farm product amounts and seasonality and examine spatial relationships between small farms and K-12 public schools in Mississippi. Online survey participation invitations were sent via email to farmers and school food service directors from October 2021 to January 2022. Data were summarized using descriptive statistics and proximities between farms (*n* = 29) and schools (*n* = 122) determined using spatial analysis. Median yearly amounts for both fresh fruits and vegetables ranged from 1–50 to 201–500 pounds while other product amounts ranged from 1–50 to >1000 pounds. Fresh fruits, fresh vegetables, and other product seasonality ranged from 1 to 6 months, 1 to 12 months, and 3 to 12 months, respectively. In total, 8 out of 12 fresh fruits, 24 out of 25 fresh vegetables, and all other products were harvested during the academic school year. Fifty percent of the schools were within a 20-mile radius of at least one small farm, while 98% were within a 50-mile radius. While many product amounts were small (1–50 pounds), most were harvested during the school year and in close proximity to at least one school. Contracting directly with farmers may be more attractive to school food authorities given current supply chain disruptions and decreasing product availability for school meal programs.

## 1. Introduction

Mississippi is one of the most agriculturally rich states in the United States (US) and remains a major supplier of domestic products such as sweet potatoes, berries, melons, and greens [1]. Farm to table initiatives, such as farm to school (F2S), have traditionally focused on purchasing from small to mid-scale farmers who need to diversify their markets to remain financially viable [2]. However, Mississippi’s largely rural land mass coupled with few centrally located resources, such as processing plants and food business centers, can make it difficult for individual producers to participate in F2S. Organizations such as the Mississippi Food Policy Council and the Mississippi Farm to School Network have worked with producers in Mississippi to strengthen and promote the locally grown food movement. Yet, limited availability of local foods and difficulty finding local producers were top challenges reported by school food authorities in the 2019 F2S Census [3]. Similarly, a lack of relationships with school staff and mismatch between produce seasonality and the academic school year were top challenges reported by producers in Mississippi [4,5].

Despite the challenges surrounding F2S activities, including local food procurement by schools, benefits to both school children and farmers are numerous. Results from a comprehensive review of F2S in the US suggest that students like having access to local foods and F2S activities may increase students’ nutritional knowledge and willingness to try fruits and vegetables [6]. Evidence also suggests that F2S programs can result in an approximate one-serving increase in fruit and vegetable consumption in school children [7]. Additionally, local food procurement by schools can contribute to the local economy and lead to diversified markets and increased off-season sales for farmers [6]. While direct sales to schools may represent a modest proportion of all farm sales (13%), they still contribute to the diversification of farm income [8]. Hence, to guide efforts for promoting and ensuring that locally grown foods are being served in Mississippi kindergarten through 12th grade (K-12) school cafeterias, more and updated knowledge about small farm product amounts and seasonality are needed. Therefore, the purpose of this study was to determine amounts and seasonality (harvest months) for products grown or raised by small farmers in Mississippi, and to examine spatial relationships between small farms and K-12 public schools in Mississippi.

## 2. Materials and Methods

The Mississippi F2S Study was a two-part online survey that was conducted with farmers (part one) and school food service directors (SFSD) (part two) in Mississippi. This study reports results about farm product amounts and harvest months from the farmer survey as well as geographic information system (GIS) mapping analysis of participating farms and schools. Primary results from the small farmers’ survey [5] and school food service directors’ survey (Thomson et al., under review) are presented elsewhere. Online data collection occurred from October 2021 through January 2022.

Mississippi farmers were identified using university extension services’ workshop/training attendee lists, United States Department of Agriculture (USDA) Agricultural Research Service scientists’ farmer contacts, farmers’ market vendor lists, and Facebook. Farms were classified based on the USDA definition of small farm as an operation with yearly gross cash farm income under USD 250,000 [9]. Farms that fit this definition were classified as small, and those that exceeded the income limit were classified as not small. Mississippi K-12 public SFSD were identified using the Mississippi Department of Education’s District Directory, school district websites, individual school websites, and telephone calls to school administrators. The study was approved and classified as exempt by the Institutional Review Board of Delta State University (protocol 21-036). All participants provided electronic informed consent before completing the surveys.

Prior to conducting the study, a literature review was performed to search for questionnaires that could be used for the present study’s purpose. No single questionnaire was found that covered all the topics of interest; hence, researchers combined and modified items from published questionnaires to create the questionnaires for this study. The majority of survey items were taken from the 2016 Nevada Department of Agriculture F2S Survey [10], including farm product amounts grown, raised, or produced per year and harvest months which have not been previously published. The six responses for farm product yearly amounts (in pounds) included 1–50 (extra small), 51–100 (small), 101–200 (medium), 201–500 (large), 501–1000 (extra large), and >1000 (extra-extra large). Although the Nevada survey was intended for farmers, items used from the questionnaire were modified for applicability to SFSD so that comparisons could be made between SFSD and farmers. Complete descriptions of survey item sources have been previously published [5] or are under journal review (Thomson et al., under review). Although demographic characteristics, including gender, age, ethnicity/race, marital status, and education, were collected on farmers and SFSD, they are not reported in this paper and may be found elsewhere [5] (Thomson et al., under review).

The questionnaires were created using Snap Desktop software, version 12.06 (Snap Surveys Ltd., Portsmouth, NH, USA). Three researchers with experience in F2S methodology, working with small farmers, or working with SFSD reviewed the questionnaire for content and face validity. Additionally, the study team completed the online questionnaire several times to test routing rules and ease of use. Based on the reviews and testing, list revisions and small wording changes were made. Invitations were sent by email using Snap Online (Snap Surveys Ltd., Portsmouth, NH, USA), an online mobile and secure survey management system. The email invitations contained both a link to the questionnaire and a link to opt out of the study. Five reminder emails were sent, approximately 2 weeks apart, to individuals who had not yet completed the questionnaire. Additionally, one research team member called individuals whose contact information included a telephone number to encourage them to take the survey. If an individual could not be reached by the fourth call, no further contact attempts were made although messages were left for those with voicemail. Individuals who submitted the questionnaire received a USD 20 e-gift card sent via email.

Statistical analyses were conducted using SAS^®^ version 9.4 (SAS Institute Inc., Cary, NC, USA). Descriptive statistics including frequencies, percentages, medians, means, and standard deviations were computed. Farm products were classified as available during the academic school year if harvest months occurred from August through May and not available if the only harvest month(s) was June or July. If a harvest month was chosen but the product yearly amount was missing, then a conservative and logical approach (i.e., producing at minimum 1 pound of product) was taken by assigning the lowest amount (1–50 pounds) to the missing values (70 out of 382; 18%).

An online geographic tool, Google Maps (https://www.google.com/maps/ (accessed on 20 April 2022)), was used to convert physical farm and school addresses to latitude and longitude coordinates. Radial buffers were generated at 20 and 50 miles around each farm to calculate the number of participating schools that fell within each of these ranges using QGIS version 3.24 (Open Source Geospatial Foundation Project). The buffers were chosen based on the 2019 F2S Census response choices for the question about school food procurement definition of local [3]. Of the 38 small farmers who completed the survey, 29 were included in the GIS analysis; 2 were excluded because they did not sell products for human consumption and 7 were excluded because their physical address was not obtainable.

## 3. Results

Response rates were 17% (43 out of 258) for farmers and 71% (122 out of 173) for school food service directors. In total, 5 of the 43 farmer participants did not fit the definition of a small farm and hence were excluded from data analysis resulting in an analytic sample size of 38 small farmers. Table 1 contains farm product yearly amounts (pounds) by frequencies of farmers in each amount category. The most frequently grown fruits were watermelon and cantaloupe, while the most frequently grown vegetables were okra, cucumbers, greens, and yellow squash. The most frequently grown or raised “other products” were herbs and beef. Yearly median amounts for both fresh fruits and fresh vegetables ranged from extra small (1–50 pounds) to large (201–500 pounds), while yearly median amounts for other products ranged from extra small (1–50 pounds) to extra-extra large (>1000 pounds). Farm products with the largest yearly median amounts included blueberries, muscadine grapes, leeks, beef, and pork.

Figure 1 illustrates farm product seasonality (harvest months) by frequencies of farmers in each amount category (e.g., XS1 = one farmer harvested 1–50 pounds of corresponding product yearly). Seasonality of fresh fruits ranged from 1 month (apples) to 6 months (cantaloupe and strawberries) with 4 out of 12 fresh fruits (apples, blueberries, peaches, and raspberries) not available during the academic school year. Seasonality of fresh vegetables ranged from 1 month (leeks) to 12 months (cabbage, greens, and radishes) with only 1 out of 25 (black eyed peas) not available during the academic school year. Seasonality of other products ranged from 3 months (pecans) to 12 months (beef, chicken, eggs, herbs, and pork) and all were harvested during the academic school year.

Half (50%, 61 out of 122) of the schools were within a 20-mile radius of at least one small farm while almost all (98%, 119 out of 122) were within a 50-mile radius of at least one small farm. The three schools not within a 50-mile radius of a small farm were all located in the upper northeast corner of Mississippi. Additionally, while approximately three-fourths of the farmers grew either fruits (72%, 21 out of 29) or vegetables (69%, 20 out of 29), over half grew both fruits and vegetables (59%, 17 out of 29), while between 14% (4 out of 29) and 24% (7 out of 29) grew fruits, vegetables, and another food (meat, eggs, herbs, or pecans).

## 4. Discussion

Results from the study indicate that Mississippi small farmers are growing or raising farm products typically in the range of 1–100 pounds per year, although larger amounts were reported for some products, particularly blueberries, black eyed peas, leeks, beef, and pork. In contrast, the most frequent amount reported by Nevada farmers in a 2016 survey was in the range of 100–200 pounds, although results were reported in aggregate for the 44 participating farmers [11]. The current study results also indicate that the majority (40 out of 45) of Mississippi small farm products are harvested during the academic school year. These results are supported by those reported in the 2016 Nevada survey for which all 17 farm products for which data were collected were harvested during the academic school year [11]. Further, all but three of the participating schools in the current study were within a 50-mile radius of at least one small farm. Taken together, these results indicate that sizable amounts of local foods of interest to Mississippi K-12 public schools are harvested during the school year and with relatively low transportation inputs. However, based on results reported in a recent Mississippi school food procurement study, feeding 500 school children at least one locally procured fruit or vegetable each week will require 20,000 pounds of greens, 6000 pounds of sweet potatoes, 3125 pounds of watermelon, and 1000 pounds of blueberries [12]. Hence, while some small farmers in the current study could supply these amounts to an average-sized Mississippi school district (e.g., blueberry farmers), other produce may require schools to purchase from multiple small farmers, food hubs, or processing plants to meet their needs.

The harvest data collected in the current study suggest that farmers’ and school food authorities’ perceptions about seasonality challenges for local food production may no longer hold true. The increased use of high tunnels by small farmers in Mississippi may at least partly explain the expanded seasonality of farm products (Dr, Christine Coker, personal communication, 19 September 2022). High tunnels protect plants from severe weather and allow farmers to begin their growing season earlier in the spring and later in the fall and sometimes even year-round [13]. Additionally, financial assistance is available from the USDA Natural Resources Conservation Service to fund high tunnels and supporting practices associated with their use making them an attractive choice for small farmers wishing to expand their growing season [14].

Contracting directly with local producers may be more attractive to school food authorities than traditional procurement methods, such as use of national chain vendors or distributors given current and possibly continuing issues with food supply chains [15] and decreasing availability of products for school meal programs [16]. While produce amounts reported by farmers in the current study are relatively small, purchasing foods in smaller quantities and considering value added products, such as washed and packed salad green mixes and processed sweet potatoes, may prove beneficial to enhancing the amount of locally grown foods served in school lunches. Likewise, having a stable market and set prices could benefit small farmers searching for more consistent or alternative markets. However, issues such as lack of connections between small farmers and school food authorities as well as transportation and delivery methods continue to challenge successful and sustainable F2S partnerships.

Resources are available to facilitate partnerships between small farmers and schools including Mississippi Market Maker, a free internet marketing tool that links growers and producers with consumers and is provided by the Mississippi State University Extension Service [17,18,19]. Additionally, the Mississippi F2S Network website contains resources pages for both farmers and schools with advice on when and how to contact and talk with one another [20]. Further, recent funding to create USDA regional food business centers, particularly in priority areas with high need and limited resources including the Delta [21], may help small farmers access new markets and take advantage of available resources, thereby possibly alleviating some challenges in production amounts, transportation, and delivery of farm products. Finally, the USDA Local Food Purchase Assistance Cooperative Agreement Program, whose purpose is to maintain and improve food and agricultural supply chain resiliency, may help build and expand economic opportunities for local and socially disadvantaged farmers in addition to increasing local food consumption in underserved communities [22].

Strengths of this study include the collection of farm product amounts and seasonality, and the determination of farm proximity to K-12 public schools. Limitations of this study include nonrepresentative samples of small farms and K-12 public schools in Mississippi. Finding and contacting small farmers can be difficult due to the nature of their work (mostly outdoors) and the lack of internet presence for some. Missing physical addresses for some small farmers further reduced the sample size and ability to determine small farms’ proximity to schools. Additionally, missing amount and harvest month data may have resulted in underrepresentation of farm products. Estimating farm product amounts and seasonality may be difficult for some farmers, particularly those who are not directly involved in off-farm sales. Hence, there may be inaccuracy in the farm product amount and seasonality data. Further farm product amounts were reported as harvested (grown, raised, or produced) and may not represent amounts available for purchase by schools (e.g., farmers selling to other markets).

## 5. Conclusions

This study contributes valuable and updated findings about Mississippi small farm products’ amounts and seasonality. Based on the study’s findings, while many farm product amounts were somewhat small, most products were harvested during the academic school year and in relatively close proximity to at least one K-12 public school. Collection of longitudinal data will be key for determining if USDA food system initiatives and programs will positively impact the economic viability of small farmers and lead to the consumption of more local foods in Mississippi communities, particularly by school children. Additionally, longitudinal data is needed to determine if barriers to farmers’ F2S participation are lessening or possibly changing and to track the proportion of farm income resulting from F2S participation.

## Figures and Tables

**Figure 1 ijerph-20-03572-f001:**
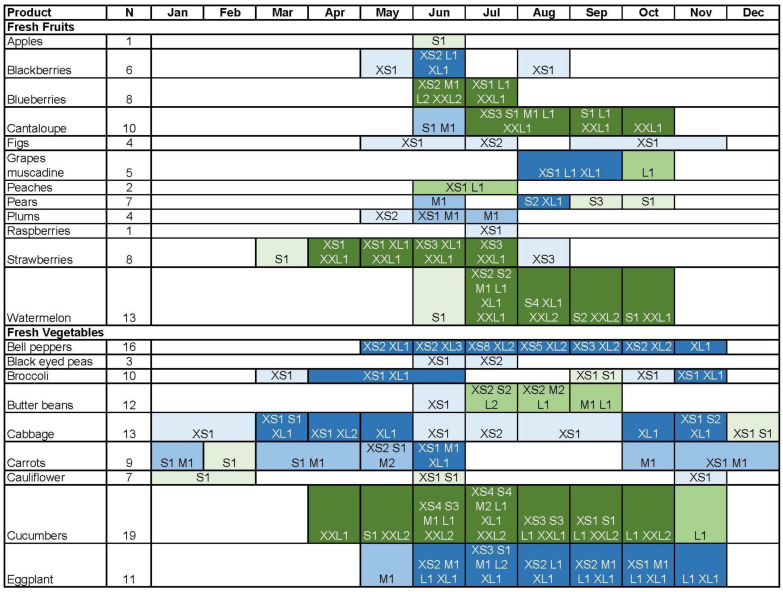
Small farm products (rows) by amount (S, M, L, XL, XXL; individual cells), harvest month (Jan–Dec; columns), and number of farmers (numbers 1–12 after amount letter; individual cells). Numbers may be less than N due to missing data or may exceed N due to multiple months chosen. Mushrooms and sweet potatoes are not included in the figure because amount and harvest month were missing. XS = extra small amount (1–50 pounds); S = small amount (51–100 pounds); M = medium amount (101–200 pounds); L = large amount (201–500 pounds); XL = extra large amount (501–1000 pounds); XXL = extra-extra large amount (>1000 pounds). Number after amount category indicates number of farmers producing product in specified amount (e.g., XS2 = 2 farmers producing product in amount of 1–50 pounds).

**Table 1 ijerph-20-03572-t001:** Small farm product yearly amounts by frequencies of farmers (*n* = 38).

		Amount Category Number and Amount (Pounds)								
		1	2	3	4	5	6			
Product ^1^	N (Missing)	1–50	51–100	101–200	201–500	501–1000	>1000	Median	Mean	SD
*Fresh fruits*										
Apples	1 (0)	0	1	0	0	0	0	2.0	2.0	-
Blackberries	6 (0)	4	0	0	1	1	0	1.0	2.2	1.8
Blueberries	8 (1)	2	0	1	2	0	2	4.0	3.6	2.1
Cantaloupe	10 (2)	4	1	1	1	0	1	1.5	2.4	1.8
Figs	4 (0)	4	0	0	0	0	0	1.0	1.0	0.0
Grapes muscadine	5 (0)	2	0	0	2	1	0	4.0	3.0	1.9
Peaches	2 (0)	1	0	0	1	0	0	2.5	2.5	2.1
Pears	7 (2)	0	3	1	0	1	0	2.0	2.8	1.3
Plums	4 (1)	2	0	1	0	0	0	1.0	1.7	1.2
Raspberries	1 (0)	1	0	0	0	0	0	1.0	1.0	-
Strawberries	8 (2)	3	1	0	0	1	1	1.5	2.7	2.3
Watermelon	13 (2)	1	5	1	1	1	2	2.0	3.2	1.8
Total farmers	-	24	11	5	8	5	6	-	-	-
*Fresh vegetables*										
Bell peppers	16 (3)	10	0	0	0	3	0	1.0	1.9	1.8
Black eyed peas	3 (1)	1	0	0	0	0	1	3.5	3.5	3.5
Broccoli	10 (1)	6	2	0	0	1	0	1.0	1.7	1.3
Butter beans	12 (3)	2	3	2	2	0	0	2.0	2.4	1.1
Cabbage	13 (3)	4	3	1	0	2	0	2.0	2.3	1.6
Carrots	9 (2)	3	1	2	0	1	0	2.0	2.3	1.5
Cauliflower	7 (4)	1	2	0	0	0	0	2.0	1.7	0.6
Cucumbers	19 (3)	5	5	2	1	1	2	2.0	2.6	1.7
Eggplant	11 (2)	4	1	1	2	1	0	2.0	2.4	1.6
Green beans	8 (2)	3	2	0	1	0	0	1.5	1.8	1.2
Green peas	3 (1)	1	0	0	1	0	0	2.5	2.5	2.1
Greens	19 (5)	5	3	1	1	3	1	2.0	2.8	1.8
Kale	11 (2)	5	1	1	2	0	0	1.0	2.0	1.3
Kohlrabi	3 (1)	1	0	0	1	0	0	2.5	2.5	2.1
Leeks	2 (1)	0	0	0	1	0	0	4.0	4.0	-
Mushrooms	1 (1)	-	-	-	-	-	-	-	-	-
Okra	20 (2)	7	6	2	1	1	1	2.0	2.2	1.5
Onions	7 (3)	2	0	0	1	1	0	2.5	2.8	2.1
Purple hull peas	18 (2)	2	5	3	3	3	0	3.0	3.0	1.4
Radishes	8 (2)	4	0	1	1	0	0	1.0	1.8	1.3
Sweet corn	10 (1)	3	3	0	1	1	1	2.0	2.7	1.9
Sweet potatoes	1 (1)	-	-	-	-	-	-	-	-	-
Swiss chard	5 (1)	3	0	0	0	1	0	1.0	2.0	2.0
Tomatoes	18 (2)	4	4	2	3	1	2	2.5	2.9	1.7
White potatoes	4 (2)	1	0	1	0	0	0	2.0	2.0	1.4
Yellow squash	19 (3)	4	3	4	2	2	1	3.0	2.9	1.6
Zucchini	15 (3)	3	4	2	0	3	0	2.0	2.7	1.6
Total farmers	-	84	48	25	24	25	9	-	-	-
*Other products*										
Beef	9 (0)	1	0	0	1	2	5	6.0	5.0	1.7
Chicken	3 (1)	1	0	0	1	0	0	2.5	2.5	2.1
Eggs	8 (0)	3	3	0	2	0	0	2.0	2.1	1.2
Herbs	10 (2)	6	0	0	0	2	0	1.0	2.0	1.9
Honey	3 (0)	0	1	1	0	0	1	3.0	3.7	2.1
Pecans	7 (0)	1	1	3	2	0	0	3.0	2.9	1.1
Pork	1 (0)	0	0	0	0	0	1	6.0	6.0	-
Total farmers	-	12	5	4	6	4	7	-	-	-

SD, standard deviation. ^1^ No farmer reported growing bunch grapes or producing cow’s milk.

## Data Availability

The data presented in this study are openly available in the USDA National Agricultural Library’s Ag Data Commons. [dataset] J.L. Thomson, T.I. Walls. 2022. Mississippi farmers’ interest in and experience with farm to school; Ag Data Commons; https://doi.org/10.15482/USDA.ADC/1527734. [dataset] J.L. Thomson, T.I. Walls. 2022. Mississippi school food service directors’ interest in and experience with farm to school; Ag Data Commons; https://doi.org/10.15482/USDA.ADC/1527826.

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
