# Peer review of "Mississippi Small Farm Product Amounts, Seasonality, and Proximity to K-12 Public Schools"

_ijerph, 2023, doi:10.3390/ijerph20043572_

Round 1
Reviewer 1 Report
Summary: This study evaluated the product yield and timing of harvest on small farms in Mississippi, and evaluated the distance these farms have from local public schools. The authors introduce the relationship between these pieces of data, however, there are several missing elements to making the connection between farm and school that need to be addressed.
Specific feedback
Title-
1. In the title and throughout the paper, the word "availability" is used. However, this is misleading to the reader. Farmers reported yield amounts, which do not indicate availability to schools- it is likely a portion of the crop is already allocated elsewhere, and thus not the entire yield is available.
Abstract-
2. Line 23: A word is missing- likely adding "proximity" to the phrase "in close to at least" will address this issue.
Introduction-
3. The introduction does a concise job of presenting evidence that the connection between farm and school could be beneficial to both, and outlining the barriers to that connection. It would be helpful to provide additional context, including areas where F2S is done well and the impact this has had on both farmers and schools, and areas where this has failed and subsequent consequences.
4. Additionally, a stronger argument could be made for the benefits of F2S. Why is it important that children are more willing to try fruits and veg? Is there data showing how much support farmers get from off-season sales?
5. The last line of the conclusion mentions "underserved communities," yet no evidence is provided in the introduction that this is the same population as that sampled.
Materials and Methods-
6. Overall this section would benefit from an organization that flowed more clearly from methods to materials. There is some back-and-forth explanation that readers may find confusing.
7. Line 80: How accurate are farmers in reporting crop yields annually?
8. Line 111: The authors note that a conservative approach to replacing missing values was taken. Is there a precedent or justification for this approach? (Additionally, for the results- I may have missed where the authors listed how many times this occurred).
Results-
9. As mentioned under the comment for the title, the authors should clarify that the results represent total yield, not surplus (amount available for use by schools).
10. Table 1- The authors could consider adding totals per product category. I was curious as to the totals for "fresh fruits," for example.
11. Figure 1- The abbreviations in this figure need further explanation. I was unable to parse the meaning from the figure caption. Similar to comment 10, adding totals per product category may also be useful here.
12. Figure 2- I have an ethical concern about the representation of the physical locations of study participants. Please clarify that both farmers and schools provided informed consent to their locations being potentially revealed in this way.
Discussion-
13. Line 173- The Materials and Methods section describes how farms were divided into small or not small based on the pounds per year reported in their yield. Does this apply differently for farmers who are producing something relatively light (e.g., blueberries) vs. something heavy (e.g., beef)? The inclusion later in that paragraph of the amount needed per school child was very helpful in providing some context. I get that a pound is a pound; however, the reflection of human hours and other resources to get a pound of beef seems different than a pound of blueberries.
14. Lines 202-3- The authors state concerns about supply chain issues and decreasing availability of products for school meal programs. Please provide evidence that these are occurring and are expected to occur going forward.
15. Line 204- The explanation of why farmers and school authorities would find F2S "attractive" needs expansion.
16. Lines 211-213- The authors note barriers to sustainable F2S partnerships. This seems like a missing piece to the study, especially given the next paragraph which outlines several resources to facilitate the partnerships. Identifying yield and proximity to schools is a basic first step that requires the reader to make the leap between using surplus yield in the school cafeteria. The study would be significantly strengthened if barriers in Mississippi specifically were identified and attempts at rectification given the strategies listed were evaluated. Are the farmers who participated aware of these resources?
17. Lines 228-238- The authors list several limitations of the study. Two notable ones that have been mentioned previously in this review are the ability of farmers to accurately report their yield and the difference between yield and availability to schools. All studies relying on self-report have some level of inaccuracy inherent in the data. However, it seems possible that the authors could re-analyze the data using actual availability (not total yield) to schools, and that this information would be more useful than total yield.
18. Lines 228-238- The authors could consider discussing the strengths of the study in addition to limitations.
Conclusion-
19. It bears noting that the reported yield of products is not sufficient to meet the needs of 500 school children (as outlined around line 186).
20. What future directions are suggested by this study?
Author Response
- Summary: This study evaluated the product yield and timing of harvest on small farms in Mississippi and evaluated the distance these farms have from local public schools. The authors introduce the relationship between these pieces of data, however, there are several missing elements to making the connection between farm and school that need to be addressed.
Response: Thank you for this comment. We have addressed specific comments and suggestions in the following items.
- Title: In the title and throughout the paper, the word "availability" is used. However, this is misleading to the reader. Farmers reported yield amounts, which do not indicate availability to schools- it is likely a portion of the crop is already allocated elsewhere, and thus not the entire yield is available.
Response: Thank you for pointing out the ambiguity in our use of “availability.” To precisely describe our data and analysis, we have replaced “availability” with “seasonality” or “harvested” in the title and throughout the manuscript.
- Abstract, Line 23: A word is missing- likely adding "proximity" to the phrase "in close to at least" will address this issue.
Response: Thank you – we have made the requested change (line 23) and deleted a word (line 24) to remain within the journal’s word limit for the abstract.
- Introduction: The introduction does a concise job of presenting evidence that the connection between farm and school could be beneficial to both and outlining the barriers to that connection. It would be helpful to provide additional context, including areas where F2S is done well and the impact this has had on both farmers and schools, and areas where this has failed and subsequent consequences.
Response: Thank you for this suggestion. Per journal instructions, we would like to keep our introduction brief rather than provide an extensive overview of F2S. We have modified a sentence in the introduction to point interested readers to a reference that provides a comprehensive review of F2S in the US (lines 46-49).
- Introduction: Additionally, a stronger argument could be made for the benefits of F2S. Why is it important that children are more willing to try fruits and veg? Is there data showing how much support farmers get from off-season sales?
Response: Thank you for these suggestions to strengthen our Introduction. We have added additional details supporting benefits of F2S for school children and farmers (lines 49-51, 53-54).
- Conclusion: The last line of the conclusion mentions "underserved communities," yet no evidence is provided in the introduction that this is the same population as that sampled.
Response: Thank you for pointing out this discrepancy. We have replaced the word “underserved” with “Mississippi” (line 259).
- Materials and Methods: Overall this section would benefit from an organization that flowed more clearly from methods to materials. There is some back-and-forth explanation that readers may find confusing.
Response: We have reviewed the Materials and Methods section and believe that the flow is appropriate. The first 2 paragraphs describe the methods, the 3rd and 4th paragraphs describe the materials, and the last 2 paragraphs describe the data analysis.
- Materials and Methods, Line 80: How accurate are farmers in reporting crop yields annually?
Response: We have added the potential for inaccuracy in the self-reported data in the limitations paragraph of the Discussion (lines 248-249).
- Materials and Methods, Line 111: The authors note that a conservative approach to replacing missing values was taken. Is there a precedent or justification for this approach? (Additionally, for the results- I may have missed where the authors listed how many times this occurred).
Response: Thank you for pointing out these missing study details. We have added the requested information (lines 116-118).
- Results: As mentioned under the comment for the title, the authors should clarify that the results represent total yield, not surplus (amount available for use by schools).
Response: Thank you for pointing out the ambiguity in our use of “availability.” To precisely describe our data and analysis, we have replaced “availability” with “seasonality” or “harvested” throughout the manuscript.
- Table 1: The authors could consider adding totals per product category. I was curious as to the totals for "fresh fruits," for example.
Response: Thank you for this suggestion. We feel it is more informative to indicate the total number of farmers for each amount category rather than an overall amount per product type because amounts are not reported as a point estimate but rather an estimated range. Hence, we feel that summing the ranges will not provide informative (meaningful) estimates.
- Figure 1: The abbreviations in this figure need further explanation. I was unable to parse the meaning from the figure caption. Similar to comment 10, adding totals per product category may also be useful here.
Response: Thank you – we have added more explanation to our figure legend (lines 154-155). Again, we feel that summing ranges of product amounts will not provide informative (meaningful) estimates.
- Figure 2: I have an ethical concern about the representation of the physical locations of study participants. Please clarify that both farmers and schools provided informed consent to their locations being potentially revealed in this way.
Response: Thank you for bringing this ethical concern to our attention. We have removed Figure 2 from our manuscript.
- Discussion, Line 173: The Materials and Methods section describes how farms were divided into small or not small based on the pounds per year reported in their yield. Does this apply differently for farmers who are producing something relatively light (e.g., blueberries) vs. something heavy (e.g., beef)? The inclusion later in that paragraph of the amount needed per school child was very helpful in providing some context. I get that a pound is a pound; however, the reflection of human hours and other resources to get a pound of beef seems different than a pound of blueberries.
Response: Thank you for this comment. We did not classify farms as small or not small based on reported yields but rather using the USDA definition that is based on gross farm income (lines 72-73). Hence, there is no difference in this classification based on the type of farm product grown, raised, or produced.
- Discussion, Lines 202-3: The authors state concerns about supply chain issues and decreasing availability of products for school meal programs. Please provide evidence that these are occurring and are expected to occur going forward.
Response: We have added a reference supporting continuing supply chain issues for food retailers (line 213, reference 15).
- Discussion, Line 204: The explanation of why farmers and school authorities would find F2S "attractive" needs expansion.
Response: Thank you – we have rearranged this sentence to clarify that continuing issues with food supply chains and decreasing availability of products are reasons why school food authorities might find contracting with local producers more attractive than contracting with traditional sources (lines 210-213).
- Discussion, Lines 211-213: The authors note barriers to sustainable F2S partnerships. This seems like a missing piece to the study, especially given the next paragraph which outlines several resources to facilitate the partnerships. Identifying yield and proximity to schools is a basic first step that requires the reader to make the leap between using surplus yield in the school cafeteria. The study would be significantly strengthened if barriers in Mississippi specifically were identified and attempts at rectification given the strategies listed were evaluated. Are the farmers who participated aware of these resources?
Response: These are excellent suggestions and as the reviewer pointed out, the study represents “first steps” in measuring the Mississippi F2S environment in terms of farmers and school food service directors (SFSD). We agree that next steps should involve rectification of barriers identified by farmers and SFSD. We did not ask the farmers in our study if they were aware of the resources listed in our manuscript so we cannot answer this question. However, it is an excellent idea for a follow-up study.
- Discussion, Lines 228-238: The authors list several limitations of the study. Two notable ones that have been mentioned previously in this review are the ability of farmers to accurately report their yield and the difference between yield and availability to schools. All studies relying on self-report have some level of inaccuracy inherent in the data. However, it seems possible that the authors could re-analyze the data using actual availability (not total yield) to schools, and that this information would be more useful than total yield.
Response: Thank you for this suggestion. While analyzing the data in terms of actual availability is an excellent idea, we did not ask the farmers in our study to identify the surplus amounts of their crops. Hence, the suggested analysis is not possible.
- Discussion, Lines 228-238: The authors could consider discussing the strengths of the study in addition to limitations.
Response: Thank you for this comment. We have added the strengths of our study to the Discussion (lines 238-239).
- Conclusion: It bears noting that the reported yield of products is not sufficient to meet the needs of 500 school children (as outlined around line 186).
Response: Thank you for this comment. We did mention that only some small farmers could supply food in the quantities needed for an average Mississippi school district in the sentence that follows the one referenced by the Reviewer (lines 196-199).
- Conclusion: What future directions are suggested by this study?
Response: We have added additional suggested future directions in the Conclusions (lines 260-262).
Reviewer 2 Report
Thank you for conducting this valuable research. The written presentation of this research is well structured and clearly represented.
Figure 1 is difficult to understand. Questions as follows 1) Why only Jan-June. Does this represent the growing season or food available season or school season? 2) Grapes, leeks (one of the times it is presented), pecans, apples, green peas, and kohlrabi, have an n but not associated data. 3) Some products are duplicated without an understanding why 4) overall question the value of this table as it is visually represented and would recommend a different visual representation be adopted.
Author Response
- Thank you for conducting this valuable research. The written presentation of this research is well structured and clearly represented.
Response: Thank you for your positive comments about our research and manuscript.
- Figure 1 is difficult to understand. Questions as follows: Why only Jan-June. Does this represent the growing season or food available season or school season?
Response: We collected data for the entire year (Jan-Dec). However, because of the size of the figure, we split the data into 2 sections – Jan-Jun and Jul-Dec.
- Figure 1: Grapes, leeks (one of the times it is presented), pecans, apples, green peas, and kohlrabi, have an n but not associated data.
Response: The data for the above-mentioned fruits and vegetables are present in the following months as shown in the figure. Apples = Jun; grapes = Aug-Oct; green peas = May-Jun; kohlrabi = Mar; leeks = Oct; pecans = Oct-Dec.
- Figure 1: Some products are duplicated without an understanding why.
Response: Because of the size of the figure, we split the data into 2 sections – Jan-Jun and Jul-Dec. Hence, the products are not duplicated but represented on both sections of the figure.
- Figure 1: Overall question the value of this table as it is visually represented and would recommend a different visual representation be adopted.
Response: We will ask the Editor if the figure can be presented in landscape orientation which will make it easier to read.